# Healthcare Service Efficiency: An Empirical Study on Healthcare Capacity in Various Counties and Cities in Taiwan

**DOI:** 10.3390/healthcare11111656

**Published:** 2023-06-05

**Authors:** Jih-Shong Wu

**Affiliations:** College of General Education, Chihlee University of Technology, New Taipei City 22050, Taiwan; jishong@mail.chihlee.edu.tw; Tel.: +886-2-2257-6167 (ext. 2316)

**Keywords:** hospital, healthcare, efficiency, data envelopment analysis (DEA), Taiwan

## Abstract

As human lifespan increases and the need for elderly care grows, the demand for healthcare services and its associated costs have surged, causing a decline in the operational efficiency of universal healthcare. This has created an imbalance in medical services across different regions, posing a long-standing challenge for the public. To address this issue, strategies to enhance the capacity, efficiency, and quality of healthcare services in various regions must be developed. The appropriate allocation of medical resources is a fundamental requirement for countries to establish a robust healthcare system. This empirical study utilized data envelopment analysis (DEA) to evaluate the efficiency of medical service capacity and identify potential improvement strategies for counties and cities in Taiwan during the period from 2015 to 2020. The results of this study show that (1) the annual average efficiency of medical service capacity in Taiwan is approximately 90%, indicating that there is still room for a 10% improvement; (2) among the six municipalities, only Taipei City has sufficient healthcare capacity, whereas the efficiency of the remaining municipalities needs improvement; and (3) most counties and cities demonstrated increasing returns to scale, indicating a need to scale up the capacity of medical services as appropriate. Based on the findings of this study, it is recommended that medical personnel be increased accordingly to balance the workload, a favorable working environment be provided to stabilize the medical workforce, and urban–rural medical disparities be balanced to improve service quality and reduce cross-regional health services. These recommendations are expected to provide a reference for society as a whole to promote and enhance public health policies, leading to a continual improvement of the quality of medical services.

## 1. Introduction

The increase in average life expectancy and demand for elderly care has led to two related challenges in healthcare systems worldwide: a surge in the demand for healthcare capacity and associated costs [1]. According to recent statistics from the United Kingdom, population aging alone is expected to result in a 3.3% surge in healthcare demand over the next 15 years [2]. With the advancement of medical technology and an increase in people’s income, there is a growing expectation that healthcare systems should offer high-quality care at lower costs. The aging population has further intensified the demand for healthcare services. As a result, governments have responded by increasing taxes to meet the rising medical needs and reducing other local expenditures to promote more efficient healthcare [3]. Using the United Kingdom as an example, in 2016, both public and private medical care expenditures accounted for almost 10% of the country’s GDP [1]. In Germany, national hospitalization expenses increased by 3.9% from EUR 87.8 billion in 2016 to EUR 91.3 billion in 2017. In light of global efforts to promote health insurance systems, healthcare expenses serve as the primary source of funds for hospital operations and survival. Consequently, medical institutions need to review and enhance the efficiency of their operating costs continually [4].

### 1.1. Healthcare in Taiwan

Taiwan has established a national health insurance system since 1995, fostering high-quality medical talents and capabilities as well as medical technology advancements over the years. Medical institutions offer services through well-trained professionals, reasonable medical expenses, and affordable health examinations, reducing the financial burden on patients. While the national health insurance system provides a comprehensive healthcare safety net, its effectiveness in providing financial protection has gradually weakened, leading to a widening gap in medical security between the rich and poor [5,6]. Specifically, since 2000, there has been a decline in the system’s provision of financial protection, while both the total medical spending and health insurance benefits for the poor have only increased at a slow rate. This phenomenon can be attributed to recent changes in the economic climate, which have resulted in negative growth in the disposable income of the poorest households. The revenue for the national health insurance program comes from employees (insured), employers, and the government. The percentage of the revenue coming from the government, the insured, and the employers was 10%, 30%, and 60%, respectively. Additionally, the increasing registration fees, copayments between 5% and 30%, and self-paid services have affected the medical utilization of the poor. Under these circumstances, the healthcare utilization of the poor has been impacted. In fact, the increase in national health insurance benefits in recent years has primarily benefited the rich rather than the poor, which was not the original intention of implementing national health insurance. To narrow the widening gap in medical security between the rich and poor, public health policies should focus on improving income disparities and promoting health insurance reforms. The ultimate goal should be to promote the health of the public [7,8]. In the 21st century, new healthcare systems must prioritize six major goals for improvement, which include safety, timeliness, effectiveness, efficiency, equity, and patient-centeredness, with “equity” aiming to ensure access to quality medical services for all [9].

The COVID-19 disease first discovered at the end of 2019 caused a global outbreak in early 2020. Taiwan was highly recognized by the global medical community for its high-quality national health insurance system and modern medical technology and techniques, which quickly controlled the pandemic and reduced the number of deaths in the early stages. However, in mid-April 2022, Taiwan faced a new wave of Omicron COVID-19 and adopted a model of normal life, balancing the economy and pandemic prevention while gradually opening the borders. Unfortunately, some counties and cities with large populations and serious COVID-19 epidemics were severely affected, with medical capacity stretched thin, especially the areas that were already under heavy demand during normal times. Moreover, due to the low medical deductibles and the convenience of seeking medical treatment under Taiwan’s national health insurance system, some waste of medical treatment resources occurred frequently. To avoid causing financial pressure on citizens, the government often implements policies to reduce medical expenses. As a result, the medical industry is competing not only with peer medical institutions but also the ever-changing trends of medical technology and the healthcare industry environment. The future trends in the medical business environment are specialization, technology, and large-scale hospitals, which will impact the survival of medical institutions and the balanced development of urban and rural medical services. Governments and health departments around the world are actively seeking ways to improve and balance the medical service capacity, equipment, and technology in different regions to meet the public’s expectations for quality and equitable medical care.

This study compiled the number of medical practitioners per 10,000 population in 19 counties and cities in Taiwan from 2015 to 2020 based on statistical data from the Ministry of Health and Welfare [10] and the Department of Household Registration, Ministry of the Interior [11], as shown in Figure 1. The 19 counties and cities included New Taipei City, Taipei City, Taoyuan City, Taichung City, Tainan City, Kaohsiung City, Yilan County, Hsinchu County, Miaoli County, Changhua County, Nantou County, Yunlin County, Chiayi County, Pingtung County, Taitung County, Hualien County, Keelung City, Hsinchu City, and Chiayi City. Figure 1 highlights the large gap in medical service capacity among counties and cities in Taiwan, with Hsinchu County, Miaoli County, and New Taipei City ranking at the bottom. Notably, New Taipei City, with a population of about 4 million people, ranks third from the bottom in terms of the number of medical practitioners per 10,000 people, indicating a significant deficiency in medical service capacity. In addition, distance to medical treatment is an important factor that affects the public’s medical utilization, with the chances of medical resources being used decreasing as distance increases [12,13,14]. Seeking medical treatment across districts reflects that people make a trade-off between distance and expected medical service and quality [15,16]. Most people in New Taipei City travel to Taipei City for medical treatment, which allows the service capacity of New Taipei City to be maintained under normal circumstances. However, the severe COVID-19 surge from April to June 2022 and the principle of avoiding cross-county or cross-city admission have led to a shortage of medical personnel and insufficient capacity, highlighting the long-standing issue of the uneven distribution of medical resources in Taiwan. Cross-region admissions and excessive workloads of healthcare workers have become the norm. Balancing and distributing medical service capacity and resources in various counties and cities have become important issues that need to be addressed by society in the future.

### 1.2. Measuring Healthcare Efficiency

There are two main research and evaluation methods used to measure efficiency and performance over the past 50 years: mathematical programming and econometric analysis. These methods fall under data envelopment analysis (DEA), which is commonly used to measure and estimate the production possibility frontier. The DEA method is recognized as an effective method for performance evaluation and benchmarking that is widely used in healthcare and many industries [17,18]. Simar and Wilson [19] advocated for DEA as the most convenient and effective measurement tool for hospital efficiency measurement [17]. DEA is primarily used to explore the relative efficiency between the multiple inputs and outputs of each decision-making unit (DMU), while stochastic frontier analysis (SFA), a representative econometric analysis method used to estimate the production frontier, explores the absolute efficiency and interrelation between multi-input items and environmental impact factors for a single production item.

In 1983, Nunamaker [20] applied DEA to healthcare for the first time, analyzing the relative efficiency of medical services in 17 hospitals. Using hospitalization expenses as the output item and three input items of the elderly and children, female patients, and other hospitalization days, Nunamaker [20] found that more than 60% of the hospitals were inefficient. Caballer-Tarazona et al. [21] investigated three medical service units in 22 hospitals in Valencia, Spain, using DEA and found that the efficiency rate of general surgery was 36%, while ophthalmology was 41% and traumatic orthopedic surgery was 27%. Kounetas and Papathanassopoulos [22] applied DEA to evaluate hospital productivity and found that more than 80% of the hospitals had a technical efficiency lower than 80%, and medical equipment allocation was inappropriate in healthcare expenditure. Stefko et al. [23] applied the DEA method to evaluate the efficiency of Slovakian hospital institutions from 2008 to 2015, including five inputs, such as hospital beds, medical personnel, medical equipment, nuclear magnetic resonance (MR), and computed tomography (CT) equipment, and two outputs, including hospital bed usage and average nursing care time. The research showed that the addition of MR, CT, and medical equipment as inputs did not significantly affect the overall efficiency of medical institutions. Luasa et al. [24] used DEA to investigate the technical and scale efficiency of 39 public and 73 private nursing homes in Ireland and found that the number of medical staff was the most important labor input, but technical efficiency was only 62%. Most private nursing homes had lower efficiency than public nursing homes, and public nursing homes should expand their operating scale to improve overall efficiency. Chen et al. [25] analyzed the impact of the 2005–2012 recession on the performance of Pennsylvania hospitals using DEA. The results showed that the average efficiency dropped to 2.43% from 3.07% after the recession, indicating a slight decline in hospital performance due to the recession. Berger et al. [26] applied DEA to investigate how the European debt crisis affected the development model of hospital efficiency. The study revealed that the annual efficiency change in hospitals in low-debt states was 1.1% lower than hospitals in high-debt states after the economic crisis. Such systematic differences were not observed before the economic crisis, indicating that external impacts on public finances may increase budgetary pressures on public finance institutions. Lin et al. [27] employed DEA to assess the operating efficiency of 19 medical centers in Taiwan. The study selected five inputs, including total hospital beds, the total number of physicians, the total number of equipment, net fixed assets, and the rate of emergency transfer inpatient stays over 48 h. The output items were surplus or deficit of appropriation, length of hospital stay, total relative value units of outpatient service, total relative value units of inpatient service, and income from self-paid services. The research results indicated that the medical centers had a technical efficiency of 96%, pure technical efficiency of 99.1%, and scale efficiency of 96.8%. Private medical centers were found to be more efficient than public medical centers. Goodarzi et al. [28] utilized DEA and SFA methods to evaluate the efficiency of Kermanshah University Medical Center between 2001 and 2007. The study found that both DEA and SFA obtained the same technical efficiency of 95%. Personnel costs represented 95% of the fixed costs, implying that reducing redundant human resources may effectively lower hospital and healthcare costs.

In conclusion, evaluating healthcare efficiency is a difficult and complex task. The DEA is a non-parametric technique that uses linear programming to estimate the frontier approach. This method can estimate the relative efficiency of multiple input and output items and is commonly used to evaluate the performance of non-profit organizations, hospitals, and other medical service providers. Fazria and Dhamanti [29] analyzed 586 research articles in 23 countries and found that the five most commonly used input elements for evaluating hospital efficiency are the number of beds, medical staff, non-medical staff, medical technicians, and operation costs. The most commonly used output elements are the number of inpatients, operations, emergency department visits, outpatients, medical costs, and the number of days of hospital stay.

Presently, most research in the field of healthcare has focused on the operating efficiency of hospitals, with little attention paid to the overall regional capacity and efficiency of medical services. In response, this study employed DEA, which is capable of evaluating multiple input and output items of medical services, to analyze the capacity and efficiency of medical services in 19 counties and cities in Taiwan. By using data on the number of outpatients and emergency department visits, inpatients, and medical personnel, this study aimed to examine and discuss the efficiency and regional differences of medical services in Taiwan. It is hoped that the findings of this study can draw the public’s attention to the importance of balancing healthcare services across different regions and can serve as a reference for policymakers in formulating public health policies aimed at improving the accessibility and quality of medical services.

## 2. Methodology

Efficiency evaluation involves estimating the operational performance of a DMU and identifying the operational and efficiency space for improvement. Farrell [30] first proposed and studied efficiency measurement methods. In his study, the target of efficiency is to measure the optimal efficiency of input and output items utilizing the maximum output or minimum cost as the purpose. He also modified the function model of Debreu [31] and Koopmans [32] to define and develop a simple method for measuring and estimating efficiency that can handle multiple inputs and outputs. Farrell [30] believed that a DMU’s efficiency consists of two parts: technical efficiency (TE) and allocative efficiency (AE). TE refers to a DMU’s ability to achieve maximum output with a certain combination of inputs, while AE refers to the optimal input ratio when the relative price is unchanged. Combining the two can estimate the total economic efficiency (TEE) of the DMU. Farrell [30] proposed that efficiency measurement and estimation can adopt either non-parametric or parametric methods. His approach has become a pioneer in recent years in measuring and estimating production boundaries, one of the trending and popular research methodologies [33]. DEA is the most commonly used non-parametric method, and SFA is the most representative parametric method, both of which are often used to evaluate various efficiencies of hospitals.

After Farrell [30] proposed a non-parametric method to evaluate technical efficiency, his research method was relatively neglected until Charnes, Cooper, and Rhodes [34] referred to his concept and proposed the DEA’s CCR model (named after Charnes, Cooper, and Rhodes), which established the theoretical foundation for non-parametric efficiency estimation. The CCR model is under the assumption of output-oriented production and Constant Returns to Scale (CRS) production. Later, Banker, Charnes, and Cooper [35] removed the CRS restriction in the CCR model and proposed the BCC (Banker, Charnes, and Cooper) model, assuming Variable Returns to Scale (VRS), where increasing some input items would not increase the output in proportion to the increase in input. Academics and researchers now widely recognize the CCR and BCC models as two of the most influential models in the DEA field [36]. Cost efficiency is also known as input-oriented total economic efficiency, which focuses on reducing the cost of input items, while revenue efficiency is output-oriented total economic efficiency that concentrates on increasing revenue from output items. This study aimed to allocate the most effective medical resources and invest the most economical medical resources to maintain and improve the existing medical service capacity in various counties and cities in Taiwan as the goal and level. To achieve this objective, the study utilized the DEA input-oriented CCR (CRS) and BCC (VRS) models as empirical research methods.

### 2.1. Input-Oriented CCR Model

The following Equation (1) shows the mathematical programming model of the input-oriented CCR model, assuming that a specific DMU has *m* inputs and *s* outputs among a total of *n* DMUs, and *h_j_* represents the efficiency index of the evaluated DMU [33]:(1)$$Max\;\;\;h_{j}=\frac{\sum_{r=1}^{s}u_{r}Y_{rj}}{\sum_{i=1}^{m}v_{i}X_{ij}}$$
$$s.t.\;\;\frac{\sum_{r=1}^{s}u_{r}Y_{rj}}{\sum_{i=1}^{m}v_{i}X_{ij}}\leq 1,\;\;\;j=1,\cdots ,n$$
$$u_{r},v_{i}\geq \varepsilon \geq 0,\;\;\;r=1,\cdots ,s,\;\;\;i=1,\cdots ,m$$

In this model, *h_j_* denotes the relative efficiency value of the *j*-th DMU, *Y_rj_* denotes the *r*-th output quantity of the *j*-th DMU, and *X_ij_* denotes the *i*-th input quantity of the *j*-th DMU. The weighting coefficients *u_r_* and *v_i_* must be positive and non-zero, while ɛ is a non-Archimedes number, typically set to 10^−4^ or 10^−6^ in practical applications, which represents the minimal positive value [33].

### 2.2. Input-Oriented BCC Model

The input-oriented BCC model differs from the CCR model by including an additional variable *u_0_*, which represents the type of scale returns, as shown in Equation (2) [33].
(2)$$Max\;\;\;h_{j}=\frac{\sum_{r=1}^{s}u_{r}Y_{rj}-u_{0}}{\sum_{i=1}^{m}v_{i}X_{ij}}$$
$$s.t.\;\;\;\frac{\sum_{r=1}^{s}u_{r}Y_{rj}-u_{0}}{\sum_{i=1}^{m}v_{i}X_{ij}}\leq 1,\;\;\;j=1,\cdots ,n$$
$$u_{r},v_{i}\geq \varepsilon \geq 0,\;\;\;r=1,\cdots ,s,\;\;\;i=1,\cdots ,m$$

Since Equation (2) is a fractional programming equation, it may have infinite solutions during the solving process. Therefore, to facilitate the solution, the equation is transformed into a linear programming model. This transformation involves limiting the denominator to 1, resulting in Equation (3) [33].
(3)$$Max\;\;\;h_{j}=\sum_{r=1}^{s}u_{r}Y_{rj}-u_{0}$$
$$s.t.\;\;\;\sum_{i=1}^{m}v_{i}X_{ij}=1$$
$$\sum_{r=1}^{s}u_{r}Y_{rj}-\sum_{i=1}^{m}v_{i}X_{ij}-u_{0}\leq 0,\;\;\;j=1,\cdots ,n$$
$$u_{r},v_{i}\geq \varepsilon \geq 0,\;\;\;r=1,\cdots ,s,\;\;\;i=1,\cdots ,m$$
where from *u*_0_, the circumstances in which returns to scale can be derived are as follows:

When the optimal solution is *u*_0_ = 0, the returns to scale are Constant Returns to Scale (CRS);

When all optimal solutions *u_0_* satisfy *u*_0_ > 0, the returns to scale are decreasing returns to scale (DRS);

When all optimal solutions *u_0_* satisfy *u*_0_ < 0, the returns to scale are increasing returns to scale (IRS) [33].

## 3. Data and Empirical Model

### 3.1. Samples and Data Sources

This study analyzed the efficiency of medical service capacity in Taiwan using four inputs and three outputs that reflect the service capacity characteristics of medical institutions. The results of the study provided suggestions and improvement strategies for enhancing the efficiency of medical service capacity for each county and city. Data used in this study were obtained from the Ministry of Health and Welfare [10] on the service capacity statistics of medical institutions, including public and private hospitals and clinics. The population data were obtained from the Global Information Network of the Department of Household Affairs of the Ministry of the Interior [11]. The sample for this study comprises data from 19 counties and cities, involves 7 input and output items, and spans a period of 6 years (2015–2020). The research data for this study included a total of three output elements, namely the number of outpatients, emergency department patients, and hospitalizations in each county and city each year, and four input elements, which were the total number of hospital beds, the number of doctors (including Western medicine, Traditional Chinese medicine, and dentistry), nursing professionals (including nurses and nurse practitioners), and other licensed medical practitioners, for a total of 114 (19 × 6) samples. Table 1 presents the statistical summary of the study variables. This study assumed that the medical services were mainly utilized by the residents of each county and city. However, it was not possible to exclude or evaluate the number of people who sought medical treatment across counties and cities. The aim of this research was to assess the efficiency of medical services provided by medical institutions in each county and city. To achieve this, the annual data were divided by the corresponding population to obtain the average efficiency of medical service capacity in each county and city.

### 3.2. Empirical Model

Cultivating the healthcare workforce requires extensive training and education over an extended period. Despite the solid training, Taiwan’s medical workforce faces a heavy burden due to factors such as limited manpower growth, attrition, and long working hours. Based on the output of the existing medical service capacity, this study assumed that each county and city can invest their medical resources in the most economical way to achieve and maintain their target and level of medical service capacity. Therefore, the DEA input-oriented CCR (CRS) and BCC (VRS) models were used to estimate the technical efficiency of medical institutions in each county and city. Scale efficiency (SE = CRSTE/VRSTE) was obtained by dividing the estimated value of CCR by BCC. The study analyzed the technical efficiency of medical institutions in various counties and cities in Taiwan over a period of six years, using three output items, including the number of outpatients, emergency department patients, and hospitalizations, and four input items, which were the total number of hospital beds, physicians, nurses, and other medical professionals. The computer software used for estimation was the free DEAP Version 2.1 software provided by Coelli [37].

## 4. Results and Discussion

Table 2 and Figure 2 present the annual average efficiency estimates for medical service capacity in 19 counties and cities in Taiwan over a six-year period (2015–2020). Table 2 indicates that Taipei City, Yilan City, Nantou County, Taitung County, Hsinchu City, and Chiayi City had a relative efficiency value of 1 for the overall technical efficiency of CCR (CRS), suggesting that these counties and cities were relatively efficient in terms of their medical service capacity. Similarly, 11 counties and cities had a relative efficiency value of 1 in the pure technical efficiency of BCC (VRS), suggesting that they were relatively efficient in terms of medical service capacity. In addition, the scale efficiency (SE) values in six counties and cities, namely Taipei City, Yilan City, Nantou County, Taitung County, Hsinchu City, and Chiayi City, were 1, indicating that these institutions had the highest medical service capacity and efficiency under the most optimal production scale (see Figure 3). Furthermore, Hsinchu County, New Taipei City, and Yunlin County ranked at the bottom in terms of scale efficiency. As shown in Figure 1, these three had nearly the lowest number of medical practitioners per 10,000 population. New Taipei City, with the largest population, land area, and aging population in Taiwan, had poor scale efficiency in medical services. Fortunately, due to its proximity to Taipei City, which had the greatest medical service capacity, New Taipei City managed to maintain its medical services. Table 2 in the “RTS” column, in which 13 counties and cities are displayed as “IRS (increasing returns to scale)”, including New Taipei City, Taoyuan City, Taichung City, Tainan City, Kaohsiung City, Hsinchu County, Miaoli County, Changhua County, Yunlin County, Chiayi County, Pingtung County, Hualien County, and Keelung City. This indicated that the medical capacity of these 13 counties and cities was increasing with scale. To improve the scale and quality of medical services, it is recommended to increase the input items as appropriate, namely the investment in medical capacity.

According to Table 2, the BCC pure technical efficiency value (VRSTE) was found to be higher than the CCR technical efficiency value (CRSTE) for each county and city, except for those with a CCR technical efficiency value of 1. This indicated that the efficiency performance of most counties and cities was limited by either scale returns or production technology. While six counties and cities reached a relative efficiency value of 1 in the CCR model, 11 counties and cities reached a value of 1 in the input-oriented BCC model, with five additional counties and cities (New Taipei City, Hsinchu County, Miaoli County, Changhua County, and Chiayi County) demonstrating pure technical efficiency due to the influence of scale efficiency. The annual average efficiency values of CRS, VRS, and SE were 0.897, 0.971, and 0.924, respectively, indicating good technical efficiency, pure technical efficiency, and scale efficiency of medical institutions in Taiwan. However, there was still room for improvement at 10.3%, 2.9%, and 7.6% in terms of technical efficiency, pure technical efficiency, and scale efficiency, respectively.

The sample counties and cities analyzed in this study had varying degrees of peer references. Specifically, Hsinchu City was referenced seven times; Yilan County six times; New Taipei City and Chiayi County both four times; Hsinchu County, Miaoli County, and Nantou County three times each; and Taitung County and Chiayi City two times each. Counties or cities with more peer references were considered to have higher robustness in their medical service capacity, providing valuable input and output for other counties and cities of similar scale to improve their own efficiency. Furthermore, according to Figure 1 and Table 2, New Taipei City ranked last in the number of medical services per 10,000 people among the six major municipalities in Taiwan and also performed poorly among the nineteen counties and cities in the study, ranking sixteenth in overall technical efficiency. Moreover, among the six municipalities directly overseen by the central government, except for Taipei City, which had an efficiency value of 1, the efficiency values of the other five municipalities were low, indicating that there was considerable room for improvement. This may be partly attributed to the fact that Taiwanese people tend to utilize inpatient medical resources in an inverted pyramid manner and are accustomed to seeking medical care in large medical centers located in other regions. Therefore, the quality of primary and regional healthcare units needs to be enhanced to change the habit of cross-regional health services among Taiwanese people. Yilan County, Taitung County, and Hualien County in eastern Taiwan all had excellent annual efficiency values. However, despite having the best number of medical services per 10,000 people and ranking third in Taiwan, Hualien County had an annual efficiency value of only about 0.83, which was relatively inferior to Yilan County and Taitung County (both with efficiency values of 1). This could be attributed to the fact that Hualien County has the only medical center, Tzu Chi Medical Center, in eastern Taiwan. Although it has a high number of healthcare personnel, many of its patients are from other regions, such as the neighboring Taitung County, which has led to a decrease in the efficiency of its medical service capacity. Moreover, Hualien County’s narrow terrain and inconvenient transportation, coupled with its limited medical resources of one medical center and two regional hospitals, pose challenges to patients with severe medical conditions who often have to travel long distances for treatment, resulting in the loss of many lives. Addressing this longstanding issue should be prioritized and requires continuous improvement in public health policies.

The use of DEA in this study presents a research limitation in that the efficiency value obtained is relative rather than absolute, and any additions or reductions in the data or range of DMUs will impact it. Furthermore, it is recommended that future studies consider the inclusion of external environmental variables. Additionally, conducting a regional segmentation analysis for medical resources in remote areas can facilitate a comparison of utilization efficiency differences among metropolitan, non-metropolitan, and remote areas. This approach has the potential to enhance healthcare resource allocation and improve medical care in remote areas.

## 5. Conclusions

This study investigated the efficiency of medical services in 19 counties and cities across Taiwan collected over a six-year period (2015–2020) for a total of 798 data sets. The study employed the input-oriented DEA method of CCR and BCC, which attributes technical inefficiency as the reason for the difference between actual and frontier outputs while ignoring the impact of random factors on output. DEA can measure multiple outputs and inputs, avoiding the need for constructing a production function to estimate efficiency. As a result, the DEA method is widely used to study hospital efficiency due to its comprehensiveness and convenience.

The empirical results of this study indicated that (1) the annual average efficiency of medical services in Taiwan was about 90%, which is considered good, but there is still a 10% margin for improvement. (2) The best medical service capacity and efficiency were found in six counties and cities, including Taipei City, Yilan City, Nantou County, Taitung County, Hsinchu City, and Chiayi City. (3) Among the six major municipalities, only Taipei City had an efficiency value of 1, while the other five had poor efficiencies, leaving significant room for improvement. (4) Thirteen counties and cities, accounting for 68% of all counties and cities, demonstrated increasing returns to scale. Therefore, increasing medical capacity can enhance the quality and efficiency of medical services in these places. (5) In most counties and cities, the BCC method produced higher efficiency values than the CCR method, indicating that the efficiency performance was limited by the scale of medical capacity.

Based on the findings of this study, the following strategies are recommended to improve the efficiency of medical services in Taiwan: (1) increase the healthcare workforce as appropriate to improve the scale, service, and quality of healthcare capacity; (2) provide a good working environment for medical staff to enhance the quality of medical services and reduce staff turnover; and (3) encourage under-resourced counties and cities to attract and incentivize the establishment of high-quality medical institutions, increasing the capacity of medical services while reducing cross-regional health services. It is expected that the findings of this study would provide useful insights for medical management and policy formulation by medical service researchers and stakeholders. Furthermore, countries with comparable health insurance systems to Taiwan can adopt these recommendations to enhance the efficiency and quality of healthcare services collaboratively.

## Figures and Tables

**Figure 1 healthcare-11-01656-f001:**
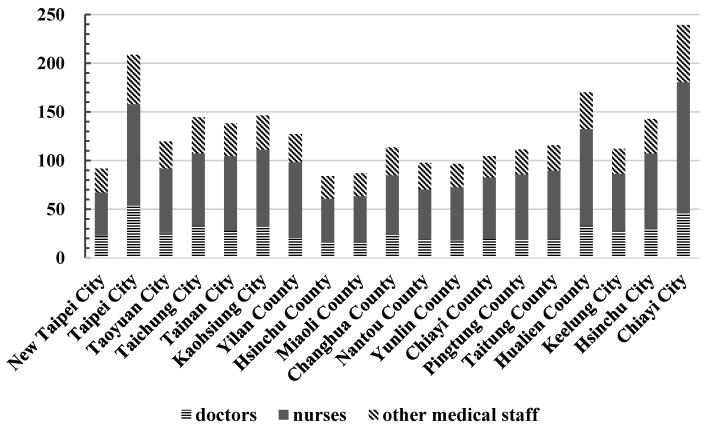
The annual number of medical practitioners per 10,000 population in each county and city from 2015 to 2020 (source data form [10]).

**Figure 2 healthcare-11-01656-f002:**
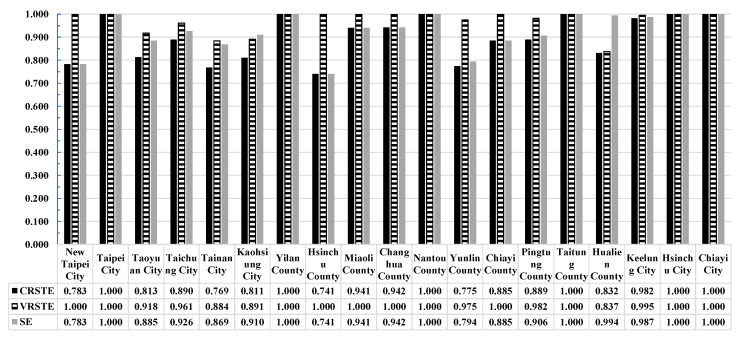
The annual average efficiency of medical service in various counties and cities in Taiwan.

**Figure 3 healthcare-11-01656-f003:**
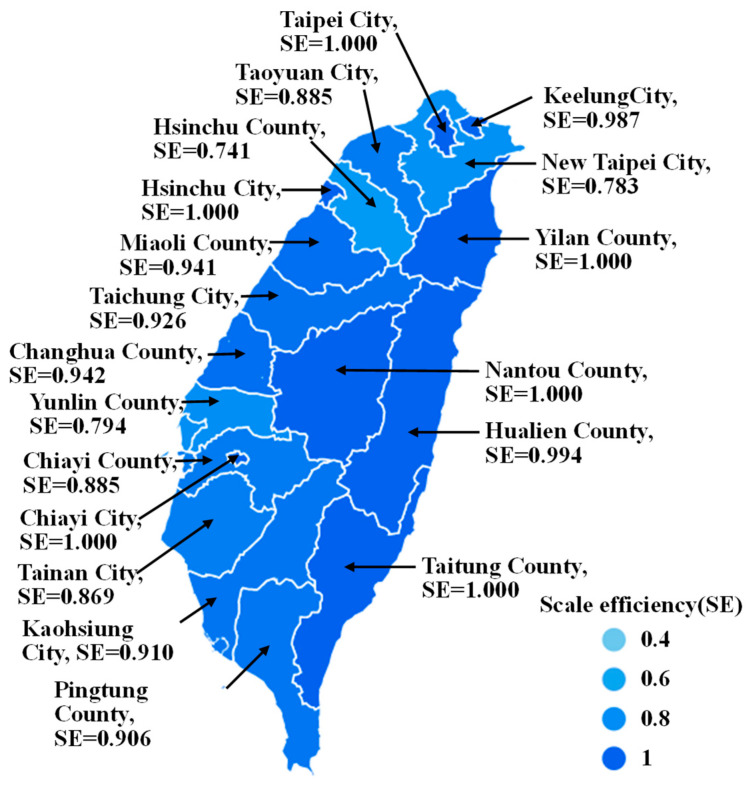
Distribution map of the annual average scale efficiency in various counties and cities in Taiwan.

**Table 1 healthcare-11-01656-t001:** Descriptive statistics of variables.

Items	Variables	Samples	Mean	Std.	Min.	Max.	Median
Output Items	Number of outpatient/year	114	5,825,277	5,937,610	734,488	24,246,518	2,555,041
	Number of emergency departments/year	114	382,517	311,772	81,936	1,121,367	203,444
	Number of hospitalizations/year	114	171,084	165,403	24,755	638,412	82,314
Input Items	Total number of beds/year	114	8682	7690	1496	25,526	4001
	Number of physicians (including Western medicine, Traditional Chinese medicine, and dentistry)/year	114	3590	3979	380	15,764	1246
	Number of nurses (nurses and nurse practitioners)/year	114	8564	8041	1430	30,091	3685
	Number of other licensed medical practitioners/year	114	3973	3975	519	15,314	1595
The population of each county and city/year			1,226,564	1,106,241	215,261	4,030,954	561,551

(Source data from [10,11]).

**Table 2 healthcare-11-01656-t002:** Estimation of the annual average efficiency of medical service capacity in various counties and cities in Taiwan.

Item	Counties or Cities (DMUs)	CCR (CRSTE)		BCC (VRSTE)		SE		RTS	Referenced County or City	Number of Times Referenced
		Efficiency Value	Rank	Efficiency Value	Rank	Efficiency Value	Rank			
1	New Taipei City	0.783	16	1.000	1	0.783	18	irs	-	4
2	Taipei City	1.000	1	1.000	1	1.000	1	-	-	-
3	Taoyuan City	0.813	14	0.918	16	0.885	14	irs	New Taipei City, Yilan County, Chiayi County, Hsinchu City	-
4	Taichung City	0.890	10	0.961	15	0.926	11	irs	Yilan County, Nantou County, Hsinchu City, Chiayi City	-
5	Tainan City	0.769	18	0.884	18	0.869	16	irs	New Taipei City, Hsinchu County, Miaoli County, Chiayi County, Hsinchu City	-
6	Kaohsiung City	0.811	15	0.891	17	0.910	12	irs	New Taipei City, Yilan County, Nantou County, Hsinchu City	-
7	Yilan County	1.000	1	1.000	1	1.000	1	-	-	6
8	Hsinchu County	0.741	19	1.000	1	0.741	19	irs	-	3
9	Miaoli County	0.941	9	1.000	1	0.941	10	irs	-	3
10	Changhua County	0.942	8	1.000	1	0.942	9	irs	-	-
11	Nantou County	1.000	1	1.000	1	1.000	1	-	-	3
12	Yunlin County	0.775	17	0.975	14	0.794	17	irs	New Taipei City, Hsinchu County, Miaoli County, Chiayi County, Taitung County, Hsinchu City	-
13	Chiayi County	0.885	12	1.000	1	0.885	14	irs	-	4
14	Pingtung County	0.889	11	0.982	13	0.906	13	irs	Yilan County, Hsinchu County, Chiayi County, Taitung County, Hsinchu City	-
15	Taitung County	1.000	1	1.000	1	1.000	1	-	-	2
16	Hualien County	0.832	13	0.837	19	0.994	7	irs	Yilan County, Chiayi City	-
17	Keelung City	0.982	7	0.995	12	0.987	8	irs	Yilan County, Miaoli County, Nantou County, Hsinchu City	-
18	Hsinchu City	1.000	1	1.000	1	1.000	1	-	-	7
19	Chiayi City	1.000	1	1.000	1	1.000	1	-	-	2
Annual average		0.897		0.971		0.924				

Note: CRSTE: constant return to scale technical efficiency; VRSTE: variable return to scale technical efficiency; SE: scale efficiency (SE = CRSTE/VRSTE); RTS: returns to scale.

## Data Availability

This data can be found here: [10,11].

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
