# Peer review of "Healthcare Service Efficiency: An Empirical Study on Healthcare Capacity in Various Counties and Cities in Taiwan"

_healthcare, 2023, doi:10.3390/healthcare11111656_

Round 1
Reviewer 1 Report
Thank you for the opportunity to review the manuscript. The manuscript presents a good evaluation of the Taiwanese health system. Please find my suggestions.
· Page 2 second paragraph; it would be more beneficial if the authors provide a detail on the health insurance scheme – for example, subscription fees (how people can join the scheme) or whether publicly funded by taxpayer money…
· Line 57 – regarding co-payments, what fraction of the cost needs to be paid?
· Line 77 and 78 please elaborate. It is not clear.
· Line 89-91. Why were the 19 counties selected? Is it the representative of the country’s counties or all the counties in Taiwan were included? Please clarify.
· The authors should remove the old citations for example, reference 20 is ok as it indicates the beginning of using DEA as evaluation tool. However, describing reference 21 and 22 is not good (lines 133-138) and should be removed.
· The authors may provide a concise summary of the use of DEA rather than explaining individual studies (Lines 130-179). This will give highlight to the subsequent paragraphs to explain the rationale of the use of DEA.
· The statistical models provided in sections 2.1 and 2.2 are difficult to understand to the readers. I would suggest (authors may not agree) to provide a summary of the models rather than giving the detailed statistical inputs for the result.
· Line 288 “…in 19 counties and cities in Taiwan” may be removed as it has been repeated few times before this.
· Line 302… why the sample is multiplied by 7 and 6. Please explain.
· Table 1. There is a great variation between the range of data in every category. Was the data skewed? Please provide an explanation to this variation ( I assume that it may be due to the population of the county). In that case, can we provide a more standardized form of data…for example per 100,000 population.
· 366… what is peer reference. Please explain.
· Lines 277-380 reference is needed to support the claim.
· Table 2, please provide the full form of abbreviations. A table should be self-explanatory. Same with figure 3
· The conclusion section is very long. It looks like a discussion. Please concise.
· The authors should explain the limitations of the study.
Author Response
Response to Reviewer 1 Comments
MANUSCRIPT TITLE:Healthcare Service Efficiency: An Empirical Study on Healthcare Capacity in Various Counties and Cities in Taiwan
Dear reviewer,
Thank you for your useful comments and suggestions on the manuscript. I have modified the manuscript to address all requested changes, and the detailed corrections are listed below point by point.
Reviewer: Thank you for the opportunity to review the manuscript. The manuscript presents a good evaluation of the Taiwanese health system. Please find my suggestions.
Response: Thank you for your comment and suggestion.
Point 1: Page 2 second paragraph; it would be more beneficial if the authors provide a detail on the health insurance scheme – for example, subscription fees (how people can join the scheme) or whether publicly funded by taxpayer money…
Response 1: Thank you for your comment. It has already been revised (Lines 59-62).
Point 2: Line 57 – regarding co-payments, what fraction of the cost needs to be paid?
Response 2: The copayments make up 5% to 30% of the total cost. This information has been added (Lines 62-63).
Point 3: Line 77 and 78 please elaborate. It is not clear.
Response 3: Thank you for your comment. This has been revised accordingly.
Point 4: Line 89-91. Why were the 19 counties selected? Is it the representative of the country’s counties or all the counties in Taiwan were included? Please clarify. Thank you for your comment and suggestion.
Response 4: There are 19 counties and cities in Taiwan.
Point 5: The authors should remove the old citations for example, reference 20 is ok as it indicates the beginning of using DEA as evaluation tool. However, describing reference 21 and 22 is not good (lines 133-138) and should be removed.
Response 5: Thank you for your comment. This has been removed accordingly.
Point 6: The authors may provide a concise summary of the use of DEA rather than explaining individual studies (Lines 130-179). This will give highlight to the subsequent paragraphs to explain the rationale of the use of DEA.
Response 6: Thank you for your comment. I have revised this section to provide a literature review focusing on the utilization of DEA to explore the input and output items involved in healthcare and hospital efficiency.
Point 7: The statistical models provided in sections 2.1 and 2.2 are difficult to understand to the readers. I would suggest (authors may not agree) to provide a summary of the models rather than giving the detailed statistical inputs for the result.
Response 7: Thank you for your comment. Sections 2.1 and 2.2 illustrate the mathematical models of this study; therefore, we believe it is important that they are retained.
Point 8: Line 288 “…in 19 counties and cities in Taiwan” may be removed as it has been repeated few times before this.
Response 8: Thank you for your comment. It has already been removed.
Point 9: Line 302… why the sample is multiplied by 7 and 6. Please explain.
Response 9: The sample for this study comprises data from 19 counties and cities, involving 7 input and output items, and spanning a period of 6 years (2015-2020).
Point 10: Table 1. There is a great variation between the range of data in every category. Was the data skewed? Please provide an explanation to this variation ( I assume that it may be due to the population of the county). In that case, can we provide a more standardized form of data…for example per 100,000 population.
Response 10: Thank you for your comment. Table 1 presents the original data for each input and output item. Towards the end of Section 3.1, we explain that the data were divided by the population of each county and city to derive the efficiency values (Lines 317-318).
Point 11: 366… what is peer reference. Please explain.
Response 11: Thank you for your comment. The peer reference pertains to a specific county or city with a higher and more superior efficiency value, serving as the reference point for other counties or cities in the peer assessment (Lines 379-381).
Point 12: Lines 277-380 reference is needed to support the claim.
Response 12: Please refer to the explanations in Response 11.
Point 13: Table 2, please provide the full form of abbreviations. A table should be self-explanatory. Same with figure 3.
Response 13: Thank you for your comment. Explanations have been included in the manuscript as notes (specifically in Lines 408-409).
Point 14: The conclusion section is very long. It looks like a discussion. Please concise.
Response 14: Thank you for your comment. This has been revised accordingly.
Point 15: The authors should explain the limitations of the study.
Response 15: Thank you for your comment. This has been revised accordingly. The limitations and recommendations of this study have been incorporated into the end of the first paragraph of the 'Conclusions' section (Lines 423-430).

Reviewer 2 Report
The subject studied in this work is current. However, the presentation of the text could be more attractive. I write some comments for the author.
ABSTRACT: I suggest adding the date of the study in the abstract.
INTRODUCTION: The introduction is too long. I suggest breaking it down into items.
DATA AND EMPIRICAL MODEL: 3.1. I think the explanation of the samples (lines 302-303) could be more understandable.
DISCUSSION: What were the limitations of the study?
CONCLUSIONS:
Conclusions are too long. The author could rewrite the main considerations.
I think the author could not conclude, “based on the findings of this study” (line 428), that health professionals' salaries should be reasonable (line 432), because this variable was not considered in this work.
Author Response
Response to Reviewer 2 Comments
MANUSCRIPT TITLE:Healthcare Service Efficiency: An Empirical Study on Healthcare Capacity in Various Counties and Cities in Taiwan
Dear reviewer,
Thank you for your useful comments and suggestions on the manuscript. I have modified the manuscript to address all requested changes, and the detailed corrections are listed below point by point.
Reviewer: The subject studied in this work is current. However, the presentation of the text could be more attractive. I write some comments for the author.
Response: Thank you for your comment and suggestion.
Point 1: ABSTRACT: I suggest adding the date of the study in the abstract.
Response 1: Thank you for your comment. I have added the date of the study in the abstract (Line 16).
Point 2: INTRODUCTION: The introduction is too long. I suggest breaking it down into items.
Response 2: Thank you for your comment. Subitems (1.1 and 1.2) have been included as recommended.
Point 3: DATA AND EMPIRICAL MODEL: 3.1. I think the explanation of the samples (lines 302-303) could be more understandable.
Response 3: Thank you for your comment.
Point 4: DISCUSSION: What were the limitations of the study?
Response 4: Thank you for your comment. This has been revised accordingly.
Point 5: Conclusions are too long. The author could rewrite the main considerations.
Response 5: Thank you for your comment. This has been revised accordingly.
Point 6: Conclusions: I think the author could not conclude, “based on the findings of this study” (line 428), that health professionals' salaries should be reasonable (line 432), because this variable was not considered in this work.
Response 6: Thank you for your comment. This has been removed accordingly (Lines 23 and 449).

Round 2
Reviewer 2 Report
Congratulations to the authors for rewriting the article briefly. However, I have two comments:
1. I think the explanation of the samples still deserves greater understanding (Lines 311-312):
“…for a total of 798 (19 x7 x 6) samples, 133 (19x 7) samples each year for six years.” What does it mean? 798 (19 counties and cities x 7 ? X 6 years) …
2. I keep thinking the conclusions are too long. There is text that should be in the section “discussion”, for example, the limitations of work.
The conclusion should include a SUMMARY of the work, RESPONDING TO THE OBJECTIVES OF THE WORK. I also suggest the inclusion of a SUMMARY of the strategies for the topic under study.
Author Response
Dear reviewer,
Thank you for your useful comments and suggestions on the manuscript. I have modified the manuscript to address all requested changes, and the detailed corrections are listed below point by point.
Reviewer: Congratulations to the authors for rewriting the article briefly. However, I have two comments:
Response: Thank you for your comment and suggestion.
Point 1: I think the explanation of the samples still deserves greater understanding (Lines 311-312):
“…for a total of 798 (19 x7 x 6) samples, 133 (19x 7) samples each year for six years.” What does it mean? 798 (19 counties and cities x 7 ? X 6 years) …
Response 1: Thank you for your comment. This has been revised accordingly. The sample for this study comprises data from 19 counties and cities, involving 7 input and output items, and spanning a period of 6 years (2015-2020). (Lines 315-318, and 324)
Point 2: I keep thinking the conclusions are too long. There is text that should be in the section “discussion”, for example, the limitations of work. The conclusion should include a SUMMARY of the work, RESPONDING TO THE OBJECTIVES OF THE WORK. I also suggest the inclusion of a SUMMARY of the strategies for the topic under study.
Response 2: Thank you for your comment. This has been revised accordingly. The limitations and recommendations of this study have been incorporated into the end of the 'Results and discussion' section (Lines 419-426). The suggested strategies has been modified in the end of the 'Conclusions' section (Lines 460-467).
